# Investigating the Regulatory Mechanism of the Sesquiterpenol Nerolidol from a Plant on Juvenile Hormone-Related Genes in the Insect *Spodoptera exigua*

**DOI:** 10.3390/ijms241713330

**Published:** 2023-08-28

**Authors:** Hanyang Dai, Baosheng Liu, Lei Yang, Yu Yao, Mengyun Liu, Wenqing Xiao, Shuai Li, Rui Ji, Yang Sun

**Affiliations:** 1Institute of Plant Protection, Jiangsu Academy of Agricultural Sciences, Nanjing 210014, China; karlclair8@163.com (H.D.); 15312093326@163.com (B.L.);; 2Key Laboratory for Conservation and Use of Important Biological Resources of Anhui Province, Anhui Provincial Key Laboratory of Molecular Enzymology and Mechanism of Major Diseases, College of Life Sciences, Anhui Normal University, Wuhu 241000, China

**Keywords:** *Spodoptera exigua*, nerolidol, transcriptome analysis, juvenile hormone esterase-like, growth and development

## Abstract

Various plant species contain terpene secondary metabolites, which disrupt insect growth and development by affecting the activity of juvenile hormone-degrading enzymes, and the juvenile hormone (JH) titers maintained in insects. Nerolidol, a natural sesquiterpenol belonging to the terpenoid group, exhibits structural similarities to insect JHs. However, the impact of nerolidol on insect growth and development, as well as its underlying molecular mechanism, remains unclear. Here, the effects of nerolidol on *Spodoptera exigua* were investigated under treatment at various sub-lethal doses (4.0 mg/mL, 1.0 mg/mL, 0.25 mg/mL). We found that a higher dose (4.0 mg/mL) of nerolidol significantly impaired the normal growth, development, and population reproduction of *S. exigua*, although a relatively lower dose (0.25 mg/mL) of nerolidol had no significant effect on this growth and development. Combined transcriptome sequencing and gene family analysis further revealed that four juvenile hormone esterase (JHE)-family genes that are involved in juvenile hormone degradation were significantly altered in *S. exigua* larvae after nerolidol treatment (4.0 mg/mL). Interestingly, the juvenile hormone esterase-like (JHEL) gene *Sexi006721*, a critical element responsive to nerolidol stress, was closely linked with the significant augmentation of JHE activity and JH titer in *S. exigua* (*R^2^* = 0.94, *p* < 0.01). Taken together, we speculate that nerolidol can function as an analog of JH by modulating the expression of the enzyme genes responsible for degrading JH, resulting in JH disorders and ultimately disrupting the development of insect larvae. This study ultimately provides a theoretical basis for the sustainable control of *S. exigua* in the field whilst proposing a new perspective for the development of novel biological pesticides.

## 1. Introduction

Throughout evolution, plants have developed intricate defense mechanisms to combat the threats posed by an array of phytophagous insects [1,2]. These strategies often involve the production of diverse secondary metabolites, which can be classified into four categories, sulfur-containing compounds, alkaloids, phenols, and terpenoids, which often display toxic or antifeedant effects on host insects [3,4,5]. Among these four phytochemicals, terpenoids have exceptional structural diversity. These compounds constitute a potent class of active substances synthesized by plants in response to stress caused by insect herbivory or mechanical damage [6,7]. Not only do terpenes act as communication molecules within plant-insect interactions, but they can also exert direct toxicity. These natural compounds play a pivotal role in plant defense [4,6,8].

Nerolidol (3,7,11-trimethyl-1,6,10-dodecatrien-3-ol) is a natural sesquiterpenol, belonging to the terpenoid group, which is found in a variety of plants, and previous studies on nerolidol have shown that it can act as a volatile signal [9,10,11]. For instance, the accumulation of nerolidol in tea plants has been linked to plant responses to mechanical damage or feeding by *Empoasca onukii*, whilst the volatilization of nerolidol could induce the resistance of other tea plants to *E. onukii* [9,10,11]. In addition, nerolidol exhibit various pharmacological and biological activities, including antioxidant, antimicrobial, antiparasitic and insecticidal activities [9]. However, there remains a paucity of research investigating the impact of nerolidol on insecticide mechanisms alongside the growth and development of insect pests.

The growth and development of insects are orchestrated by two key hormones: Ecdysterone (20E) and juvenile hormone (JH). JH is a sesquiterpene that performs critical functions in insect development, metamorphosis, and reproduction [12,13,14,15]. Several previous studies have demonstrated that sesquiterpenes obtained from plants, which share structural similarities with JH, can cause disturbances in JH-regulated physiology [4,16]. For example, three natural sesquiterpenes extracted from *Centaurea paui* (Compositae) affect the molting and metamorphosis of *Locusta migratoria* nymphs by disturbing JH biosynthesis [16]. Additionally, (E)-farnesene (EβF) is a crucial secondary metabolite found in several plants that offers indirect protection against aphids [4,17,18]. Its structure is similar to that of JH. EβF has been found to disturb the normal function of JH by upregulating key degradation genes before suppressing the expression of SeVg and SeVgR genes, resulting in a decrease in the *S. exigua* population [4]. Considering that nerolidol is structurally similar to EβF and JH, we hypothesized that nerolidol affects the growth and development of pests by disturbing JH biosynthesis.

Juvenile hormone esterase (JHE) is a member of the carboxyesterase family that functions as a juvenile hormone-degrading enzyme in the hemolymph of insects and the cytoplasm of some tissues [19]. JHE is the most significant JH-degrading enzyme and plays a vital role in controlling JH titers [19]. Furthermore, juvenile hormone esterase-like (JHEL) is another enzyme within the same gene family that shows high structural similarity to JHE, which contains the GXSAG motif in gene sequences—a hallmark of JHE, although they differ in some other conservative structures (Appendix A). Researchers have previously confirmed that JHELs also play a crucial role in the growth and development of a variety of insects, primarily by regulating larval molting and metamorphosis [20,21,22,23,24].

The beet armyworm, *Spodoptera exigua*, is a globally recognized agricultural pest with a wide-ranging appetite including over 170 crops such as economically important species from the Cruciferae, Legume, and Convolvolaceae families [25]. With its potent combination of robust migratory patterns, impressive reproductive abilities, and intermittent outbreaks, this insect routinely inflicts considerable damage to agricultural outputs [4,26]. As such, chemical control has become the primary method for curbing the havoc resulting from *S. exigua* in the field [27]. However, due to long-term unreasonable use, this pest has exhibited alarming resilience with varying degrees of resistance to a broad spectrum of chemical pesticides, including organochlorine, organophosphorus, carbamate, pyrethroid and benzoylurea [27,28,29,30]. Considering the need to mitigate pesticide residues and combat the resistance of target pests, a sustainable approach for field pest control and a cornerstone of green agricultural development would involve screening natural active substances that could serve as pioneering compounds. In a previous experiment, nerolidol was successfully identified from *Medicago sativa* Linn volatiles (unpublished data). Considering that extracting these compounds directly from plant materials is difficult for meeting the demanded amount [31], we conducted various bioinformatics and molecular biology experiments with a pure purchased compound to investigate the physiological and molecular responses of *S. exigua* to nerolidol treatment in advance. Therefore, this study could pave the way for identifying natural compounds to manage *S. exigua* in the field and potentially redefine our approach towards eco-friendly pest control.

## 2. Results

### 2.1. Toxicity of Nerolidol to S. exigua Larvae

The results of the bioassay showed that the susceptibility of *S. exigua* to nerolidol decreased with the increase in instar. For example, compared with the 1st-instar larvae, the susceptibility of 3rd-instar larvae to nerolidol decreased by 4.53 (19.48/4.30) times (Table 1). In addition, the sub-lethal doses of LC_5_, LC_20_, and LC_50_ for 1st-instar larvae with nerolidol were 0.23, 0.96, and 4.30 mg/mL, respectively (Table 1).

### 2.2. Effects of Different Sub-Lethal Doses of Nerolidol on Population Life Table Parameters of S. exigua

The results in Table 2 show that a relatively lower sub-lethal dose (0.25 mg/mL) of nerolidol had no significant effect on growth and development although a higher sub-lethal dose (4.0 mg/mL) of nerolidol significantly impaired the normal growth, development, and population reproduction of *S. exigua*. Although there was no significant difference in egg hatching rates between the groups, the intrinsic growth rate of the *S. exigua* population treated with 1.0 mg/mL and 4.0 mg/mL doses of nerolidol was significantly lower than that of the control (*p* < 0.05, Table 2). Conversely, the larval mortality of the treated population (1.0 mg/mL and 4.0 mg/mL) was significantly higher than that of the control population (*p* < 0.05, Table 2).

The findings revealed that a higher dose (4.0 mg/mL) had a profound effect on the normal growth, development, and reproductive capacity of *S. exigua*. In addition to egg hatching rates, other parameters of the *S. exigua* population treated with 4.0 mg/mL doses of nerolidol were significantly different from those of the control group. As depicted in Table 2, the developmental period of the *S. exigua* population treated with 4.0 mg/mL nerolidol was significantly extended compared to the control (*p* < 0.05), particularly in terms of the larval phase, which lasted 1.17 (16.27/13.86) times longer than that of the control population. In addition, the application of 4.0 mg/mL nerolidol significantly inhibited larval survival and oviposition in adult *S. exigua*, and larval mortality in the treated population was 6.53 (77.08/11.81) times higher than that in the control population, while fecundity per female was significantly reduced by 2.18 (468.33/215.00) times compared to the normal population (*p* < 0.05).

Thus, the larvae of *S. exigua* were treated with a higher dose (4.0 mg/mL) to study the effects of sub-lethal doses of nerolidol on different instar stages of *S. exigua* larvae. Compared to the control group, the treated larvae displayed a significantly prolonged development duration only during the 1st-instar stage (*p* < 0.01, Figure 1). However, the mortality rates at all instar stages were significantly increased (*p* < 0.05, Figure 1).

### 2.3. Effects of Nerolidol on Transcriptomes of S. exigua Larvae

The RNA libraries were prepared and sequenced, resulting in 128.19 million clean reads, with 89.20% being successfully mapped to the *S. exigua* genome. Gene expression levels were evaluated by transforming read counts to FPKM. Differentially expressed genes (DEG) analysis was then performed using a threshold of Q value (adjusted *p*-value) ≤ 0.05 and |log_2_Foldchange| ≥ 1. We identified 810 genes that exhibited differential expression in larvae treated with nerolidol compared to the negative-control ethanol group. Among these, 489 were upregulated, whilst 321 were downregulated (Figure 2).

To validate the accuracy of the transcriptome data, ten random genes, including five target genes, were selected based on annotation information and then subjected to RT-qPCR analysis (Appendix A). The results of the RT-qPCR experiments confirmed the expression patterns of these genes, thus providing further support for the reliability and accuracy of the transcriptome data.

### 2.4. Function Analysis of DEGs Involved in Nerolidol Exposure of S. exigua

To explore the possible biological effects of nerolidol exposure on beet armyworms, we performed Gene Ontology (GO) and Kyoto Encyclopedia of Genes and Genomes (KEGG) enrichment analyses for these 810 DEGs, specifically focusing on the juvenile hormone esterase-family genes in *S. exigua*. The results of this are shown in Figure 3, and the most significantly enriched GO term was found to be juvenile hormone esterase activity (Figure 3A). In terms of KEGG pathway enrichment (Figure 3B), four signaling pathways were identified: steroid biosynthesis, fat digestion and absorption, glycerolipid metabolism, and drug metabolism. Notably, no significant correlation was observed between the insect hormone biosynthesis pathway and juvenile hormone esterase-family genes under nerolidol treatment. This suggested that these genes may not be directly involved in insect hormone synthesis, which includes JH. Instead, they likely play a role in metabolic pathways.

### 2.5. Structural and Sequence Analyses of JHE-Family Genes in S. exigua

Within the *S. exigua* genome sequence, we analyzed a set of nine JHE-family genes that showed similarities to JHEs from various insects. Among these, the predicted *S. exigua Sexi020049* and *Sexi019925* showed the best match (BLASTP) to the *Heliothis virescens* JHE based on the UniProt database information, which provides high-quality annotation for proteins. Similar to all lepidopteran JHEs in this list, it has a GQSAG motif, which is a critical element of the catalytic site. *Sexi020049* encodes a 563 amino acid polypeptide, while *Sexi019925* encodes a 588 amino acid polypeptide (Appendix A, Appendix A). The remaining seven sequences in the JHE family also contained the GXSAG motif, which is a hallmark of JHE. Additionally, they exhibited common motifs such as RF, DQ, E, or GxxHxxD, lending further credence to their classification as JHEs (Appendix A, Appendix A).

### 2.6. Expression Profiles of JHE-Family Genes in S. exigua Response to Nerolidol

To determine the influence of nerolidol exposure on the expression of JHEs and JHELs, we investigated the expression profiles and responses of JHE-family genes to nerolidol. In total, nine *JHEs* or *JEHLs* were identified (Appendix A, Appendix A), most of which had wide expression profiles (Figure 4), other than *Sexi006258*, which was consistent with their fundamental and essential roles in biological functions. Interestingly, several *JHEs* showed significant changes in expression levels in larvae when subjected to nerolidol compared to the control condition. Notably, *Sexi006721* showed an 8.1-fold upregulation, whereas *Sexi018773* exhibited a 2.8 fold upregulation, with both reaching relatively high levels of expression after nerolidol exposure. These findings suggested that nerolidol exposure had had a notable effect on the expression of certain JHE-family genes, potentially influencing their biological functions and regulatory roles in response to nerolidol.

### 2.7. Detection of Target Gene Expression and Quantification of JH, JHE

We further examined the potential effect of nerolidol on the activity of JHE, which is encoded by target genes in *S. exigua* larvae, together with JH titer, which is closely related to JHE activity. As shown in Figure 5A,B, JH titer and JHE activity in *S. exigua* larvae increased gradually with the progression of the instar stages, reaching their peaks during the 4th instar and subsequently declining during the 5th instar. Notably, when the larvae were treated with nerolidol, JHE activity and JH titer both increased significantly at each instar stage compared to what occurred in the control group (Mann–Whitney U-test, *p* < 0.001). Furthermore, as shown in Figure 5C, among the four JHEL genes analyzed, only *Sexi006721* was upregulated during every age stage when subjected to the same dose of nerolidol. This indicated that this gene may respond to stress induced by nerolidol exposure, implying a significant role in the growth and development of *S. exigua* larvae.

### 2.8. Pearson Correlation Analysis

In order to assess the linear correlations between target gene expression, protein expression, and key life table parameters before and after treatment with nerolidol, we performed a Pearson correlation analysis. A strong linear correlation was indicated by a Pearson regression coefficient of >0.8, corresponding to an *R^2^* value exceeding 0.64.

First of all, the experiment verified the relationship between the transcriptional level of each target gene and its protein expression level. Table 3 shows that in the absence of nerolidol, there was no strong correlation between the expression of the four target genes and the enzyme activity of JHE or the titer of JH. However, when treated with nerolidol, only the expression of *Sexi006721* was significantly positively correlated with JHE activity (*R^2^* > 0.64, *p* < 0.001) and JH titer (*R^2^* > 0.64, *p* < 0.001). In contrast, the expression of *Sexi018733* was positively correlated with the activity of JHE (*R^2^* > 0.64, *p* < 0.001). Simultaneously, we found that JH titer and JHE activity showed a significant positive correlation without nerolidol treatment (*R^2^* = 0.76 > 0.64, *p* < 0.001), with this correlation strengthening after nerolidol treatment (*R^2^* = 0.96 > 0.64, *p* < 0.001).

Secondly, the relationship between gene expression, protein expression, and their phenotypes is discussed here. When larval mortality and developmental duration were combined, the results showed that there was a significant negative correlation between larval mortality and JH titer in the normal population (*R^2^* > 0.64, *p* < 0.001, Table 3), as well as with JHE activity (*R^2^* > 0.64, *p* < 0.001, Table 3). However, these correlations decreased significantly after treatment with nerolidol, with only larval mortality being negatively correlated with the expression of *Sexi018733* (*R^2^* > 0.64, *p* < 0.05).

In addition, there was a significant correlation between larval mortality and developmental duration after nerolidol treatment (*R^2^* > 0.64, *p* = 0.05, Table 3). These findings provide insight into the association between different variables and the impact of nerolidol treatment on these relationships.

## 3. Discussion

Plants have evolved various defense mechanisms to protect themselves from pests. One important defense strategy is the production of plant secondary substances, such as terpenoids, phenols, alkaloids, tannins, flavonoids, and steroids, which form the basis of a plant’s chemical defensive barrier [32,33]. These secondary compounds play crucial ecological roles and, thus, have been studied extensively. They can induce or deter insects from feeding on host plants, stimulate or inhibit insect feeding, and hinder insect growth and development, whilst some compounds even exhibit direct toxicity against target pests [4,18,33,34]. For example, indole has been found to be toxic to lepidopteran larvae [35], whereas monoterpenols, α-terpineol, and linalool have shown insecticidal activity against *Drosophila melanogaster* [36]. The sesquiterpene farnesene is also used as an aphid repellent [37]. In our study, nerolidol, a plant compound, was identified in the volatiles of *Medicago sativa* (unpublished data). In contrast, preliminary experiments had indicated its antifeedant effect on the target pest, *S. exigua*. Therefore, it is important to investigate the interaction between nerolidol and its host insect, *S. exigua*. In this study, we initially determined the activity of nerolidol against the 1st-, 2nd-, and 3rd-instar larvae of *S. exigua*. The lethal concentration that caused 50% mortality (LC_50_) for the 3rd-instar larvae was found to be 19.48 mg/mL. For comparison, Zuo et al. [38] tested the toxicity of 3rd-instar insects of the same strain using various conventional chemical pesticides. We found that the direct use of nerolidol as a botanical insecticide in the field may not achieve the desired effect and could increase application costs. In this study, we used sub-lethal doses of nerolidol to treat *S. exigua*. By studying the effects of this secondary compound on the growth and development of the target pest, we aimed to elucidate its toxicity mechanisms and explore its potential applications.

Herein, we used three sub-lethal concentrations (0.25 mg/mL, 1.0 mg/mL, and 4.0 mg/mL) of nerolidol to treat populations of *S. exigua*. We compared the growth and development of the treated population with that of the control population before calculating life table parameters. The results subsequently revealed that when *S. exigua* was treated with a relatively lower dose (0.25 mg/mL) of nerolidol, the growth and development of the treated population did not differ significantly from that of the control population. The tolerance of target pests to exogenous compounds can be attributed to their various detoxification mechanisms. However, when we increased the sub-lethal concentration to the 4.0 mg/mL dose and continuously fed it to the larvae throughout their entire developmental stage, the growth, development, and reproduction of the treated population were all significantly affected; in particular, the mortality of each larval instar increased significantly when *S. exigua* was treated with nerolidol (*p* < 0.05, Figure 1). And the intrinsic growth rate of the treated population was only 0.31 (0.045/0.145) times that of the control population. In addition, after nerolidol treatment, the correlations between larval mortality and JH titer and JHE activity were greatly reduced, indicating that the continuous intake of nerolidol had disrupted the hormone levels in *S. exigua* larvae whilst also significantly affecting their normal development (Figure 5, Table 3). Previous studies have also shown that when *S. exigua* specimens are treated with sub-lethal doses of another plant secondary metabolite, EβF, this significantly affects the survival rate of beet armyworm larvae and the fecundity per adult female, resulting in a significant decrease in the intrinsic growth rate of the population (*p* < 0.05) [4]. Moreover, several recent studies have demonstrated that azadirachtin, another secondary plant substance, could affect the pupariation and eclosion of *P. xylostella*, ultimately leading to the inability to engage in normal reproduction [39].

JHE is an important juvenile hormone-degrading enzyme that facilitates the critical transition from the larval to the adult stage in insects and other arthropods. This is achieved by catalyzing the ester cleavage of JH [40,41,42,43]. The seminal work of Sparks et al. [44] previously demonstrated that inhibiting JHE leads to a decrease in the JH degradation rate, resulting in the development of abnormally large larvae and delayed pupation. In this study, we used RNA sequencing (RNA-Seq) to investigate the transcriptome of *S. exigua* and observed significant alterations in the expression of four JHE-family genes (*Sexi021849*, *Sexi018733*, *Sexi006721*, and *Sexi006258*) in response to the influence of nerolidol (Appendix A). However, classic JHEs possess distinguishing motifs such as RF, DQ, E, and GxxHxxD/E. Of particular significance is the fact that the GxSxG consensus sequence encircling the serine in the active site is typically represented as GQSAG, and this sequence is instrumental in distinguishing *JHEs* from *JHELs* [22,45,46]. Upon the structural identification of the JHE-family genes discovered in this study, these four genes exhibiting significant variation in expression under nerolidol stress were classified as *JHELs*. They deviated from typical JHEs in terms of their specific conserved sequences (Appendix A, Appendix A). Furthermore, subsequent analysis of spatiotemporal expression revealed that *Sexi021849*, *Sexi018733*, and *Sexi006721* exhibited broad expression profiles across various developmental stages and distinct adult tissues. This suggests that these three JHEL genes are involved in numerous processes underlying the growth and development of *S. exigua*. Concurrently, under nerolidol stress, the expression levels of *Sexi018733* and *Sexi006721* were significantly upregulated by 2.8 and 8.1 times, respectively. This highlights the pivotal roles of these genes, as they are prominently induced by nerolidol.

In our study, newly hatched *S. exigua* larvae were consistently fed a sub-lethal dose (4.0 mg/mL) of nerolidol. Samples were obtained, and JH titer and JHE activity in the larvae were evaluated before each instar molt. Simultaneously, the expression changes in four *JHELs* (*Sexi021849*, *Sexi018733*, *Sexi006721*, and *Sexi006258*) were tracked. The analysis subsequently demonstrated a positive correlation between the expression of *Sexi018733* and *Sexi006721* and the activity of JHE in different-instar larvae (*R^2^* > 0.64, *p* < 0.01) (Figure 5, Table 3). The expression of *Sexi006721* was significantly upregulated in each instar compared to the control (Figure 5), indicating its crucial roles in the nerolidol in the response of *S. exigua* to nerolidol-induced stress and in the regulation of its growth and development. However, both *Sexi018733* and *Sexi006721* were classified as genes encoding JHELs, possibly diverging in functionality from typical JHEs. As previously demonstrated by Leboeuf et al. [24] in *Camponotus floridanus*, JHEL-family genes exhibit various functionalities beyond the direct degradation of JH, potentially extending to digestion or detoxification, independent of JH or even JH protection. Furthermore, Kontogiannatos et al. [23] discovered that the expression of JHE analog gene in *Sesamia nonagrioides* could be influenced by ecdysteroids, but not JH analog, suggesting a potential involvement in larval molting. *Sexi018733* and *Sexi006721,* detected in this study, responded to nerolidol stress and were closely linked to an increase in larval JHE activity. Sequence analysis also indicated that these two genes were closely related and potentially shared similar functions (Appendix A), but the expression of *Sexi006721* showed a higher correlation with variations in JHE activity and JH titer (*R^2^* = 0.94) (Figure 5). In addition, most studies show that JHE is imperative for initiating insect metamorphosis by reducing JH titer. Prior to larval molting or metamorphosis onset, JHE activity increases whilst JH biosynthesis declines [47,48,49]. Interestingly, in this study, under continuous nerolidol stress, while the activity of JHE increased, the titer of JH also significantly increased compared with the control, and there was a significant and strong positive correlation between them (*R^2^* > 0.64, *p* < 0.05). A pronounced elevation of the JH titer in all instar stages, especially the 5th instar stage, was sufficient for delaying larval molting and metamorphosis processes, and this led to a significant extension of larval duration (Figure 1 and Figure 5, Table 2). It is speculated that the elevation in JH titer is attributed to the continuous accumulation of nerolidol, a JH analog, or is an insect strategy to maintain hormonal balance in response to the rising JHE activity. In conclusion, the persistent exposure to nerolidol leads to hormone imbalances in target pests, which ultimately affects their growth and development. However, in fact, as a natural sesquiterpenol, the impact of nerolidol on *S. exigua* still needs to be verified using extracts obtained from plant materials. Analyzing the expression profiles of different compounds in natural volatiles, combined with the study of odorant-binding proteins (OBPs) in target pests, may help us obtain more and purer target compounds from plants [50,51].

In this experiment, transcriptome analysis provided us with the overall changes in gene expression in target pests under the stress of a plant sesquiterpenol, nerolidol, which can be a new candidate to act as a botanical insecticide. Then, spatiotemporal expression and gene family analysis are helpful for us to understand the basic information on and functions of target genes. Combined with the expression detection of target proteins, it is more helpful to locate key genes. Life table and correlation analysis further explain the role and influence of key target genes on phenotype. To sum up, applying various bioinformatics and molecular biology techniques is helpful for us to efficiently and accurately target key genes, and it also provides a basis for further functional verification. As shown in this study, the identification and interpretation of the key JHEL gene (*Sexi006721*) in responsive to nerolidol stress was important, and its expression strongly correlated with that of JHE activity and JH titer. Based on these findings, we speculate that nerolidol can act as a JH analog by affecting the expression of JHE-family genes, resulting in hormone disorders and ultimately disrupting the development of insects. However, the interaction mechanism of *Sexi006721* with JHE and JH needs to be further verified. This study provides insights into the toxicological mechanism of nerolidol against its target pest, *S. exigua,* and offers a novel perspective on the sustainable control of field pests.

## 4. Conclusions 

To sum up, in this experiment, transcriptome analysis provides us with the overall changes in gene expression in target pests under the stress of nerolidol. Then, spatiotemporal expression and gene family analysis are helpful for us to understand the basic information on, and functions of, target genes. Combined with the expression detection of target proteins, it is more helpful to locate key genes. Life table and correlation analysis further explain the role and influence of key target genes on phenotype. In summary, we found that a relatively higher sub-lethal dose (4.0 mg/mL) of nerolidol significantly impaired the normal growth, development, and population reproduction of *S. exigua.* We also carried out the identification and interpretation of the key JHEL gene (*Sexi006721*) in responsive to nerolidol stress, and its expression strongly correlated with that of JHE activity and JH titer. Based on these findings, we speculate that nerolidol can act as a JH analog by affecting the expression of JHE-family genes, resulting in hormone disorders and ultimately disrupting development in insects. However, the functions of *Sexi006721* require further verifying though other molecular biology approaches, like the CRISPR/Cas9 technique or in vitro protein expressing. Our study offers insights into the toxicological mechanism of a plant sesquiterpenol, nerolidol, which can be a new candidate to act as a botanical insecticide against its target pest, *S. exigua*. Additionally, it provides a novel perspective on the sustainable control of field pests.

## 5. Materials and Methods

### 5.1. Insect Strains

The WHS strain, a susceptible strain of *S. exigua*, was provided by Dr. Zuo Yayun of Northwest Agriculture and Forestry University in China. This strain was collected in 1998 from Wuhan, Hubei Province, and has been maintained in a laboratory without exposure to insecticides. During the larval stage, *S. exigua* were fed an artificial diet composed primarily of wheat and soybean flour, as reported by Zuo et al. [52]. The indoor feeding conditions were: temperature 26 ± 1 °C, relative humidity 60%, photoperiod 16 (Light): 8 (Dark). Upon reaching adulthood, the beet armyworms were housed in cages (30 × 40 × 50 cm) and fed a diet containing 10% sugar water. White paper was also provided in the cages for oviposition, with eggs being collected daily.

### 5.2. Chemicals and Bioassays

Cis-nerolidol used in this study was procured from Sigma-Aldrich (St. Louis, MO, USA). To determine the sub-lethal dose of nerolidol for subsequent investigations, a bioassay was conducted on 1st-, 2nd-, and 3rd-instar larvae of *S. exigua* using the method of pharmaceutical allocation previously described by Zuo et al. [38]. This involved diluting nerolidol with 95% ethanol and distilled water at seven different concentrations, with ethanol treatments being used as the controls. The 95% ethanol content was adjusted for each gradient to determine the volume of nerolidol at the highest gradient concentration.

The prepared solution was then thoroughly mixed with the liquid artificial diet, which had been previously cooled to 50 °C after high-temperature (121 °C) sterilization. Equal amounts (1.5 mL) were dispensed into each 24-well plate using a pipette. After cooling, one *S. exigua* larva was added to each well. After seven days, the mortality across all treatments were recorded. The established criterion for larval death was a lack of motion when propelled with a brush.

### 5.3. Construction of a Life Table for S. exigua under Sub-Lethal Nerolidol Treatment

Following the bioassay results, the LC_5_, LC_20_, and LC_50_ values for nerolidol against 1st-instar larvae of *S. exigua* were found to have been approximately 4.0, 1.0, and 0.25 mg/mL, respectively. These three concentrations were subsequently selected for nerolidol treatment, with a solvent control group containing only 95% ethanol being established. Throughout the entire larval stage, *S. exigua* were fed an artificial diet with sub-lethal doses of nerolidol, with each treatment containing 1200 larvae. Upon reaching adulthood, they were provided with sugar water, while the feeding conditions remained constant, as detailed in Section 5.1.

The collection and calculation of life table parameters followed the protocol previously described by Sun et al. [53]. Larvae subjected to different treatments were reared in 60 mm Petri dishes, with five larvae being housed per dish until the 4th instar, after which the density was reduced to two larvae per dish. Daily observations were performed to monitor larval growth and mortality rates until pupation. At the pupal stage, individuals were identified as male or female, with the pupal development periods and emergence rates being documented. Adult pairs (ten in total) were randomly chosen and reared in 350 mL plastic cups. The paper within these cups was replaced daily, and the number of eggs laid per female was recorded. Upon hatching, the larvae were counted, with the hatching rate being calculated. After gathering the data, the intrinsic growth rate was calculated to evaluate the effect of sub-lethal nerolidol dosage on the growth and development of *S. exigua*.

### 5.4. Sample Collection, RNA Isolation, and Library Preparation for RNA-Seq

To investigate the effect of nerolidol on *S. exigua* and identify the key genes associated with growth and development, we conducted RNA-Seq analysis of *S. exigua* larvae treated with a sub-lethal nerolidol dosage (4.0 mg/mL, corresponding to the LC_50_ of 1st-instar larvae). Ethanol-treated larvae served as controls. Treatments were initiated with newly hatched larvae, with sampling being carried out during the second-instar stage. Five larvae from each treatment group were pooled to form one biological replicate, with each treatment consisting of three replicates.

Total RNA was extracted from *S. exigua* samples using the SV Total RNA Isolation Kit (Promega, Madison, WI, USA) in accordance with the manufacturer’s instructions. Subsequently, the integrity and concentration of total RNA were determined using a NanoDrop and an Agilent 2100 bioanalyzer (Thermo Fisher Scientific, Waltham, MA, USA). RNA-Seq and gene expression analyses were conducted on an Illumina platform (Nanjing Genepioneer Biotechnologies Inc., Nanjing, China). Spatiotemporal expression RNA-Seq data were acquired from the National Center for Biotechnology Information Sequence Read Archive (accession no. PRJNA723689) for further analyses.

### 5.5. RNA-Seq Data Processing

Raw short-reads were initially curated into clean data by removing sequencing adapters and low-quality bases by the Trimmomatic software (v0.36) [54] using the main parameter “HEADCROP:20 SLIDINGWINDOW:4:15 MINLEN:50”. Clean data were mapped to the *S. exigua* reference genome [29] using HiSAT2 (v2.21) [55]. The genome-alignment SAM files were transformed into sorted-binary BAM files using SAMtools (v1.12) [56]. Subsequently, StringTie software (v2.1.4) was used to assemble potential transcripts and estimate their abundance [57]. Furthermore, a local Python script was used to generate CSV files containing count matrices for genes and transcripts derived from the Stringtie results. In addition, a heat map was constructed using Pheatmap (R package v1.0.12), with FPKM (fragments per kilobase per million mapped reads) acting as the index for evaluating gene expression levels [58]. Differentially expressed gene (DEG) analysis was performed using DESeq2 software (R package v1.37.4). All candidate DEGs were filtered based on threshold values, with a Q value (adjusted *p*-value) of ≥0.05 and an absolute |log_2_Foldchange| of ≥1, in accordance with the methodology of Love et al. [59]. To visualize the results, a volcano plot was subsequently generated using the ggplot2 R package (v3.3.5) [60].

### 5.6. GO, KEGG Pathway Enrichment Analyses, and Gene Functional Annotation

GO and KEGG pathway enrichment analyses were performed using the enricher function from the clusterProfiler R package (v3.18.1) as previously described by Yu et al. [61]. Gene function was annotated using various databases, including Nr (NCBI non-redundant protein sequences), Nt (NCBI non-redundant nucleotide sequences), Pfam (protein family), KOG/COG (Clusters of Orthologous Groups of Proteins), Swiss-Prot (a manually annotated and reviewed protein sequence database), KO (KEGG Ortholog database), and GO.

### 5.7. Quantitative Reverse Transcriptase PCR Verification

RNA isolation was performed as described previously in Section 5.4. For cDNA library synthesis, MMLV Reverse Transcriptase (Promega) treated with ribonuclease H (TCytoscapeckara, Tokyo, Japan) was used. The concentration of cDNA was quantified by using spectrophotometry. Furthermore, all primers were designed using Primer 5.0 software. For quantitative reverse transcription PCR (RT-qPCR) using SYBR Green I, primers were designed based on the genes identified by RNA-Seq analysis. Additionally, the *S. exigua* housekeeping gene, β-actin (GenBank accession no. JN616391), was used as an endogenous reference for data normalization. Appendix A provides a list of primers used for β-actin and the other ten random sequence primers used to verify the differential transcriptome sequence results.

The SYBR Premix Ex Taq Kit (Takara, Tokyo, Japan) was used for RT-qPCR, which was performed using a Bio-Rad iCycler Real-Time Quantitative RT-PCR Detection System (Bio-Rad, Hercules, CA, USA). Melting curve analysis and gel electrophoresis were performed to determine the specificity of amplified products. The RT-qPCR reactions for each treatment were replicated three times, with non-template control reactions being performed in triplicate for each primer pair. The expression levels of each gene in *S. exigua* under the different treatments were determined using the relative quantification (RQ) values, calculated using the 2^−ΔΔCt^ method [62]. Significant differences in gene expression were detected using Duncan’s multiple-range method.

### 5.8. Identification of Putative JHE and JHEL Genes in S. exigua

To identify genes encoding JHE and JHEL in the *S. exigua* genome assemblies (NCBI GCA_011316535.1), we performed TBLASTN searches (1 × 10^−5^) using *JHEs* from various species, including *Helicoverpa zea*, *Bombyx mori*, *Nilaparvata lugens*, *Tribolium castaneum*, *Thrips palmi*, *Diachasma alloeum*, *Colaphellus bowringi*, *Bemisia tabaci*, *Mythimna separata*, *Apis mellifera*, *Drosophila melanogaster*, *Culex quinquefasciatus*, *H. virescens*, *Manduca sexta*, *Choristoneura fumiferana*, and *Psacothea hilaris* as queries. This was followed by iterative searches using the newly identified *S. exigua* JHEs or JHELs as queries.

The presence of a signal peptide was predicted using SignalP 5.0 program (http://www.cbs. dtu.dk/services/SignalP/, accessed on 4 May 2023). Molecular mass and isoelectric point (pI) calculations were performed using the ExPASy Proteomics server (http://web.expasy.org/protparam/, accessed on 6 May 2023). Multiple-sequence alignment of 17 candidate proteins and 9 *S. exigua* JHE-family genes was carried out using MAFFT software to identify the seven conserved motifs (RF, DQ, GQSAG, GxxHxxD, R/Kx(6)R/KxxxR, E and T) proposed by Kamita and Hammock [63].

### 5.9. JH and JHE Assays

For the JH and JHE assays, whole *S. exigua* larvae were sampled at the end of each instar (prior to molting). Three replicates were performed here, consisting of three larvae per replicate. The samples were then homogenized using 0.01 mol/L PBS (pH = 7.0) on ice, with the supernatant being collected after centrifugation at 3000 rpm and 4 °C for 20 min.

JH and JHE detection kits (Nanjing JC Detect Biotechnologies Co., Ltd., Nanjing, China) were used to determine the levels of JE titer and JHE activity in the samples using the double-antibody sandwich method, following the manufacturer’s instructions. Absorbance (OD) was measured at 450 nm using a spectrophotometer, with the activity concentrations of JH and JHE in the samples being calculated using a standard curve.

### 5.10. Data and Statistical Analysis

Bioassay data were analyzed using probit analysis with PoloPlus software (LeOra Software 2002, Berkeley, CA, USA). Statistical analyses were performed using SPSS statistical software v12.0 (SPSS, Inc., Chicago, IL, USA).

The Mann–Whitney U-test was used to identify both the variations in developmental duration and mortality in *S. exigua* larvae under nerolidol treatment as well as the differences in JH titer and JHE activity before and after nerolidol treatment. Life table data are presented as means ± standard deviation and were analyzed using one-way analysis of variance (ANOVA), followed by Tukey’s honest significant difference (HSD) tests. Pearson correlation analysis was also employed to evaluate the relationship between key life history parameters in *S. exigua* and target gene expression, JH, and JHE levels at different doses of nerolidol.

## Figures and Tables

**Figure 1 ijms-24-13330-f001:**
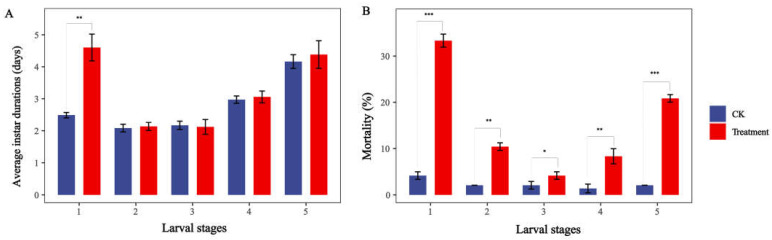
Effects of nerolidol treatment on developmental duration and mortality of *S. exigua* larvae. (**A**) Developmental duration; (**B**) mortality. Asterisks indicate significant differences: * *p* < 0.05; ** *p* < 0.01; *** *p* < 0.001 (Mann–Whitney U test).

**Figure 2 ijms-24-13330-f002:**
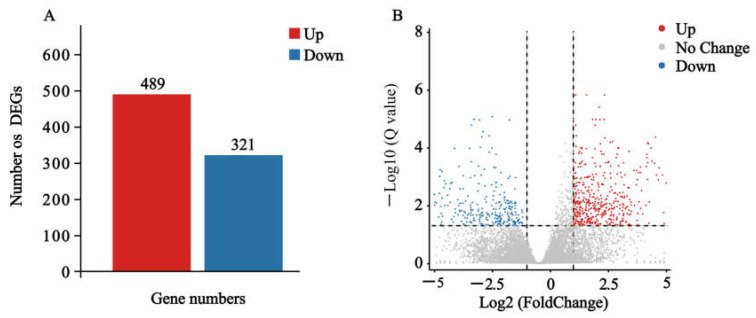
Analysis of differentially expressed genes (DEGs) in *S. exigua* larvae after nerolidol treatment. (**A**) Number of DEGs treated with nerolidol compared to the negative-control; (**B**) Volcano plot of DEGs identified in larvae.

**Figure 3 ijms-24-13330-f003:**
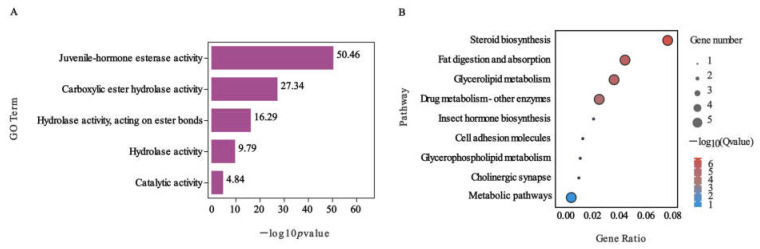
GO and KEGG analysis of JHE-family genes in *S. exigua* under nerolidol treatment. (**A**) The most significantly enriched GO term; (**B**) KEGG pathway enrichment.

**Figure 4 ijms-24-13330-f004:**
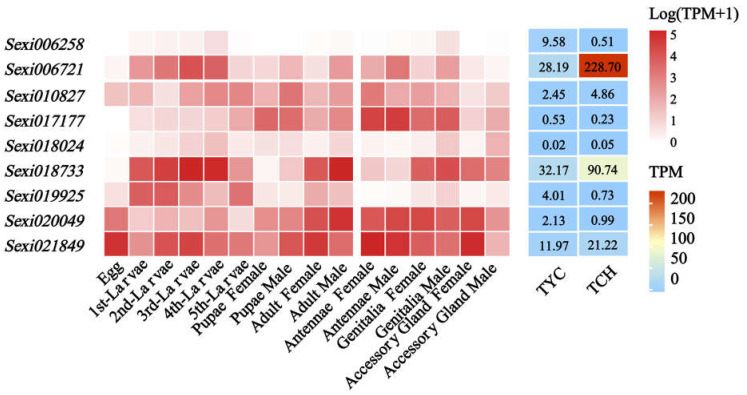
Expression profiles of *JHEs* and *JHELs* in *S. exigua* in response to nerolidol.

**Figure 5 ijms-24-13330-f005:**
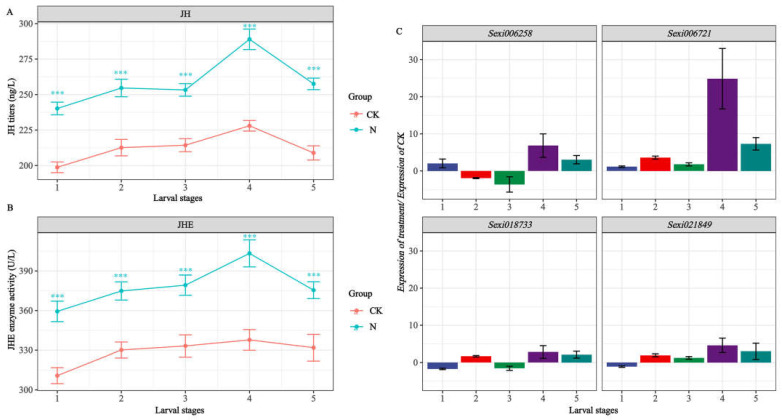
Changes in JH titer (**A**), JHE activity (**B**), and expression of *JHELs* (**C**) in different-instar larvae of *S. exigua.* treated with nerolidol. Asterisks indicate significant differences: *** *p* < 0.001 (Mann–Whitney U test).

**Table 1 ijms-24-13330-t001:** Toxicity of nerolidol to the 1st-, 2nd-, and 3rd-instar larvae of *S.exigua*.

Larval Stage (Instar)	LC_5_ (95% FL ^a^) (mg/mL)	LC_20_ (95% FL) (mg/mL)	LC_50_ (95% FL) (mg/mL)	Slope ± SE
1	0.23 (0.07–0.46)	0.96 (0.48–1.52)	4.30 (2.98–6.20)	1.29 ± 0.18
2	3.46 (0.45–6.27)	6.35 (1.97–10.41)	12.01 (6.79–24.65)	3.04 ± 0.31
3	6.72 (3.85–9.29)	11.30 (7.87–14.45)	19.48 (15.31–25.00)	3.56 ± 0.36

^a^ FL: fiducial limit.

**Table 2 ijms-24-13330-t002:** Life table parameters of *S.exigua* treated with different doses of nerolidol.

Key Life History Parameters	Different Treatments			
	CK	0.25 mg/mL	1.0 mg/mL	4.0 mg/mL
Larval stage (d)	13.86 ± 0.63 ^a^	14.45 ± 0.79 ^ab^	15.26 ± 1.21 ^ab^	16.27 ± 1.46 ^b^
Pupa stage (d)	7.45 ± 0.44 ^a^	7.97 ± 0.44 ^ab^	9.02 ± 0.65 ^bc^	10.36 ± 0.97 ^c^
Adult stage (d)	10.15 ± 0.47 ^a^	9.79 ± 0.60 ^ab^	9.30 ± 0.19 ^b^	9.05 ± 0.48 ^b^
Egg stage (d)	2.03 ± 0.12 ^a^	2.11 ± 0.13 ^a^	2.23 ± 0.26 ^ab^	2.59 ± 0.13 ^b^
The whole life-span of *S. exigua* (d)	33.49 ± 0.74 ^a^	34.31 ± 1.02 ^a^	35.81 ± 1.94 ^ab^	38.27 ± 2.75 ^b^
Larvae mortality (%)	11.81 ± 5.20 ^a^	22.22 ± 8.56 ^a^	39.58 ± 7.80 ^b^	77.08 ± 9.47 ^c^
Adult emergence rate (%)	87.20 ± 0.53 ^a^	81.10 ± 6.67 ^ab^	78.17 ± 4.60 ^bc^	75.60 ± 3.45 ^bc^
Per female fecundity	468.33 ± 60.00 ^a^	472.67 ± 24.93 ^a^	435.00 ± 34.56 ^a^	215.00 ± 30.80 ^b^
Hatching rate of eggs (%)	74.40 ± 2.65 ^a^	72.63 ± 4.40 ^a^	68.70 ± 5.42 ^a^	65.07 ± 6.71 ^a^
Intrinsic rate of increase (R)	0.145 ± 0.007 ^a^	0.133 ± 0.010 ^a^	0.110 ± 0.011 ^b^	0.045 ± 0.008 ^c^

Note: Means in the same row followed with the same letter are not significantly different at the 5% level (Tukey’s HSD tests).

**Table 3 ijms-24-13330-t003:** The Pearson correlation between respective key parameters related to JH and JHE in *S. exigua*.

Parameters	Other Parameters	Linear Equation	*R^2^*	*p*	Linear Equation	*R^2^*	*p*
		Untreated			Treated with Nerolidol		
The expression of *Sexi021849*	JH titer	y = 296.350x + 205.720	0.11	<0.001	y = 289.910x + 242.010	0.60	<0.001
	JHE activity	y = 550.150x + 316.100	0.40	<0.001	y = 239.270x + 364.480	0.53	<0.001
	Larval stage	y = 42.166x + 1.806	0.35	<0.001	y = 4.202x + 3.014	0.03	<0.001
	Larvae mortality	y = −48.893x + 3.485	0.31	0.001	y = −65.941x + 19.265	0.07	0.019
The expression of *Sexi006258*	JH titer	y = −20.623x + 212.610	<0.01	<0.001	y = −3062.300x + 265.610	0.09	<0.001
	JHE activity	y = 293.600x + 327.710	0.02	<0.001	y = −2563.800x + 384.030	0.09	<0.001
	Larval stage	y = −138.510x + 3.263	0.51	<0.001	y = −328.730x + 3.976	0.25	<0.001
	Larvae mortality	y = −34.240x + 2.482	0.02	0.001	y = −688.860x + 16.919	0.01	0.019
The expression of *Sexi006721*	JH titer	y = 2.088x + 212.510	<0.01	<0.001	y = 350.070x + 238.690	0.94	< 0.001
	JHE activity	y = 131.390x + 327.100	0.02	<0.001	y = 307.450x + 360.660	0.94	< 0.001
	Larval stage	y = −60.755x + 3.534	0.50	<0.001	y = −9.661x + 3.818	0.17	<0.001
	Larvae mortality	y = −6.817x + 2.446	<0.01	0.001	y = −147.710x + 23.958	0.40	0.019
The expression of *Sexi018733*	JH titer	y = 747.690x + 209.160	0.06	<0.001	y = 4682.900x + 230.850	0.61	<0.001
	JHE activity	y = 1270.300x + 323.020	0.17	<0.001	y = 4458.600x + 351.710	0.72	< 0.001
	Larval stage	y = −119.170x + 3.312	0.22	<0.001	y = −275.090x + 4.909	0.48	<0.001
	Larvae mortality	y = −106.110x + 2.839	0.12	0.001	y = −3452.900x + 36.119	0.78	0.019
JH titer	JHE activity	y = 0.862x + 145.580	0.76	< 0.001	y = 0.858x + 156.220	0.96	< 0.001
	Larval stage	y = 0.002x + 2.375	<0.01	<0.001	y = −0.017x + 7.535	0.06	<0.001
	Larvae mortality	y = −0.089x + 21.174	0.79	< 0.001	y = −0.361x + 108.820	0.31	<0.001
JHE activity	Larval stage	y = 0.018x − 3.014	0.05	<0.001	y = −0.031x + 14.879	0.18	<0.001
	Larvae mortality	y = −0.100x + 35.334	0.99	< 0.001	y = −0.522x + 212.950	0.50	<0.001
Larval stage	Larvae mortality	y = −0.255x + 3.070	0.04	0.513	y = 8.817x − 13.321	0.80	0.050

When the Pearson correlation coefficient was greater than 0.8, that is, when *R^2^* was greater than 0.64, there was a strong linear correlation between the two groups of data, which is marked with an underscore in the table (Table 3).

## Data Availability

The RNA-Seq data supporting the results of this article are available in the NCBI’s SRA with the accession numbers SRS10213028, SRS10213029, SRS10213030, SRR24282495, SRR24282494, and SRR24282493 (http://www.ncbi.nlm.nih.gov/sra/, accessed on 8 June 2023).

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
