# Peer review of "Investigating the Regulatory Mechanism of the Sesquiterpenol Nerolidol from a Plant on Juvenile Hormone-Related Genes in the Insect *Spodoptera exigua"

_ijms, 2023, doi:10.3390/ijms241713330_

Round 1
Reviewer 1 Report
The authors report a study on the action of nerolidol on the growth and development of Spodoptera exigua.
The manuscript is well written and very clear. The authors report a careful molecular analysis on the expression of genes involved in multiple pathways. The statistical analysis is correct and well done.
However, the authors should better emphasize the potential applications of nerolidol. Are they using a pure purchased compound? So the authors suggest purifying it from more plants? In this regard, I suggest adding the reference below and commenting on it:
Variation in Terpene profiles of thymus vulgaris in water deficit stress response, DOI: 10.3390/molecules25051091
Minor editing of English language required
Reviewer 2 Report
The article entitled: “Investigating the regulatory mechanism of the sesquiterpenol nerolidol from a plant on juvenile hormone-related genes in the insect Spodoptera exigua” is a research article in which we can find information about the effect/effects of secondary plant metabolite – nerolidol on activity of juvenile hormone, activity of juvenile hormone esterase and juvenile hormon esterase-like enzyme, an resulting changes in reproduction and development parameters of Spodoptera exigua.
Generally the srticle is understandably written, and the figures and tables are prepared well. However I have few questions and remarks.
Abstract should be carefully checked. The information about tested concentration needs to added. Two first sentences needs to be somehow linked with each other. Genes encoding specific proteins are expressed, not the proteins themselves. Please read the text carefully and spot any errors.
Line 24: which involved in juvenile hormon -> which are involved
Line 46: Nerolidol (3,7,11-trimethyl-1,6,10-dodecatrien-3-ol) is a natural sesquiterpenol that is found in a variety of plants, whilst previous studies on nerolidol have shown that it
can act as a volatile signal [9-11]. -> divide into two separate sentences or change the word “whilst”
Line 52: neuroflorins? Could you explain that? I do not uderstand that word.
Line 53: neuranol? Did you mean the substance which possess some of neuro- activity? If yes, I can not agree with that sentence
Line 74: you mean structural or functional similarity? It would be nice to add a figure with coparision of structure.
Line 89: I do not understand what is has in common with proposed article
Line 104 and M&M: in text, several times, you are using words like low, high. However it is relative term. It should be clearly written in M&M section which concentration is low, medium or hight OR changed in whole manuscript into numeral values.
Table 2: letters which indicate statistical significance might be written as power. It would be more visible and easier to compare
Line 115: highest/high or numeral value as mentioned above
Line 116: delete “normal”
Line 171 and Fig. 3: pancreatic secretion? I am guessing it was taken straight from the analysis of transcriptomic data and the assigned function/place of secretion, but it does not make any sense here, as insects do not have a pancreas. This data should be analyzed with greater deliberation, reflecting on their biological meaning
Fig. 5B: in horizontal axis word “instar” should be added. ALSO the scale is not the same which gives false impression
Line 205: here I do not understand if you were analizing activity of JHE and JHEL or only JHE and compare it with JH? It should be clarified here because JH does not possess any enzymatic activity \.
Line 219: it was a priori assumption that correlation will be linear? Clarify
Line 227: here you also mention about enzymatic activity of JH
Line 262: virulence is not a good word here as it mean ability of pathogens or microroganisms to cause demages to host organism
Line 267: the same as above
Line 281: nerol?
Line 324: “However, both Sexi018733 and Sexi006721 were classified as JHELs, possibly diverging in functionality from typical JHEs” – were classified as genes encoding JHELs
Line 348: it is not a new plant sesquiterpenol
Line 384: cooled to 50 Celsius degree?
Line 454: by “the resulting cDNA” you mean purity or concentration or something different?
Table S1: add the column with lenghth of product
In whole text some editorial mistakes were made, I just wrote some e.g. line 19, 31, 101, 192, 311, 312, 370.
Author Response
Response to Reviewer 2 Comments
Dear reviewer, thanks for your careful and professional remarks which greatly improved the quality of our research achievements, we are very happy to edit the manuscript further based on your helpful comments. Additionally, we provided point-by-point responses to your comments in the follow to outline the changes we have made, we also highlighted the changes within our revised manuscript by using the ”Track Changes” mode. We wish that our corrections are acceptable. The manuscript has also been polished by specialists in our field who are native English speakers to improve the language expression, we will appreciate everything you have done for us to improve our manuscript.
Point 1: Abstract should be carefully checked. The information about tested concentration needs to added. Two first sentences needs to be somehow linked with each other. Genes encoding specific proteins are expressed, not the proteins themselves. Please read the text carefully and spot any errors.
Response 1: Thanks for your useful suggestions, we have carefully checked the abstract and corrected some errors, as follows:
- We have added detailed concentrations in the article “under treatment at various sub-lethal doses (4.0 mg/mL, 1.0 mg/mL, 0.25 mg/mL)”, please check it (Line 18-19).
- We added a sentence “belonging to terpenoid”to make the link between the two first sentences closer, please check it (Line 16).
- We are aware of inaccuracies in “protein”and “gene” expression, in Abstract (Line 14-15), we correct the sentence “interfering with the expression of juvenile hormone-degrading enzymes, affecting the juvenile hormone (JH) titers maintained in insects” to “affecting the activity of juvenile hormone-degrading enzymes, and the juvenile hormone (JH) titers maintained in insects”; besides, in Abstract (Line 29), we add the word “genes ”, and we have also made multiple revisions in the entire text.
- We corrected some other inaccuracies in Abstractand highlighted the changes, such as add “(R2 = 0.94, P < 0.01)” (Line 28), please check it in our new submitted manuscript.
Point 2: Line 24: which involved in juvenile hormon -> which are involved.
Response 2: Thanks for your kindly reminder, we have made modifications to this sentence, please check it (Line 24).
Point 3: Line 46: Nerolidol (3,7,11-trimethyl-1,6,10-dodecatrien-3-ol) is a natural sesquiterpenol that is found in a variety of plants, whilst previous studies on nerolidol have shown that it can act as a volatile signal [9-11]. -> divide into two separate sentences or change the word “whilst”.
Response 3: Thanks for your good suggestions, we change the word“whilst”to ”and”, please check it (Line 48).
Point 4: Line 52: neuroflorins? Could you explain that? I do not uderstand that word.
Response 4: Thanks for your useful suggestions, this is a typo in our English writing and editing, we have rechecked the entire text and unified the wording of “Nerolidol”, please check it in our new submitted manuscript (Line 52).
Point 5: Line 53: neuranol? Did you mean the substance which possess some of neuro-activity? If yes, I can not agree with that sentence.
Response 5: Thanks for your useful suggestions, same as Response 4, we corrected it, please check it (Line 55).
Point 6: Line 74: you mean structural or functional similarity? It would be nice to add a figure with coparision of structure.
Response 6: Thanks for your good suggestion, this is indeed an unclear expression. Actually, we mean structural similarity, the structural similarities and differences between JHE and JHEL are shown in Table S1, and we have also added relevant expressions in the text “which contained the GXSAG motif in gene sequences, a hallmark of JHE”, please check it in our new submitted manuscript (Line 77-78).
Point 7: Line 89: I do not understand what is has in common with proposed article.
Response 7: Thanks for your good suggestion, actually, we would like to express the strong level of resistance of beet armyworm population in the field to chemical pesticides, which makes us urgently seek new and green plant-based insecticides. We have rephrased the sentence, please check it in our new submitted manuscript (Line 88-91).
Point 8: Line 104 and M&M: in text, several times, you are using words like low, high. However it is relative term. It should be clearly written in M&M section which concentration is low, medium or hight OR changed in whole manuscript into numeral values.
Response 8: Thanks for your useful suggestion, we agree with your advice and correct the relevant wording into numeral values in whole manuscript, please check it (Line 20-21, 113, 289).
Point 9: Table 2: letters which indicate statistical significance might be written as power. It would be more visible and easier to compare.
Response 9: Thanks for your useful comments, we have corrected it, please check it in our new submitted manuscript (Table 2).
Point 10: Line 115: highest/high or numeral value as mentioned above.
Response 10: Thanks for your useful suggestion, as answered in Response 8, we use numeral value in the entire text, please check it (Line 115).
Point 11: Line 116: delete “normal”
Response 11: Thanks for your kindly reminder, we delete “normal” and rephrased as “lower than that of control”, please check it in our new submitted manuscript (Line 119).
Point 12: Line 171 and Fig. 3: pancreatic secretion? I am guessing it was taken straight from the analysis of transcriptomic data and the assigned function/place of secretion, but it does not make any sense here, as insects do not have a pancreas. This data should be analyzed with greater deliberation, reflecting on their biological meaning
Response 12: Thanks for your useful suggestions, this is an oversight in our work, and we have already deleted this pathway in the new manuscript, please check it (Line 179-181, Fig. 3). Your meticulous and professional comments have prompted us to invest more enthusiasm and rigor in our subsequent research and manuscript writing process.
Point 13: Fig. 5B: in horizontal axis word “instar” should be added. ALSO the scale is not the same which gives false impression
Response 13: Thanks for your useful suggestions, we have made corrections to this figure, please check it in our new submitted manuscript (Fig. 5).
Point 14: Line 205: here I do not understand if you were analizing activity of JHE and JHEL or only JHE and compare it with JH? It should be clarified here because JH does not possess any enzymatic activity \.
Response 14: Thanks for your good suggestion, we confused the concepts of JH and JHE during the manuscript writing. And we have corrected the expressions and charts in all text, please check it in our new submitted manuscript (Section 2.7, Table 3).
Point 15: Line 219: it was a priori assumption that correlation will be linear? Clarify
Response 15: Thanks for your useful suggestions, this expression is not rigorous, we obtained the linear correlation after conducting Pearson correlation analysis, so we corrected some inaccurate expression in the original text, please check it in our new submitted manuscript (Line 232, 235, 238).
Point 16: Line 227: here you also mention about enzymatic activity of JH
Response 16: Thanks for your good suggestion, this is our mistake, we have corrected it same as Response 14, please check it in our new submitted manuscript (Section 2.8)
Point 17: Line 262: virulence is not a good word here as it mean ability of pathogens or microroganisms to cause demages to host organism
Response 17: Thanks for your useful suggestion, we have changed the word “virulence” to “toxicity”, please check it in our new submitted manuscript (Line 278).
Point 18: Line 267: the same as above
Response 18: Thanks for your useful suggestion, we have corrected it, please check it (Line 284).
Point 19: Line 281: nerol?
Response 19: Thanks for your kindly reminder, same as Response 4, we corrected it to “nerolidol ”, please check it (Line 298).
Point 20: Line 324: “However, both Sexi018733 and Sexi006721 were classified as JHELs, possibly diverging in functionality from typical JHEs” – were classified as genes encoding JHELs
Response 20: Thanks for your good suggestions, this modification makes our expression more rigorous, and we have corrected it, please check it in our new submitted manuscript (Line 341-342).
Point 21: Line 348: it is not a new plant sesquiterpenol
Response 21: Thanks for your useful suggestion, we originally intended to express that nerolidol is a new candidate botanical insecticide, new expression were given in our new submitted manuscript, please check it (Line 373).
Point 22: Line 384: cooled to 50 Celsius degree?
Response 22: Thanks for your kindly reminder, we confirm that the method is correct. Usually, artificial feed needs to be sterilized at high temperatures before use. Excessive temperatures can have an impact on nerolidol, while low temperatures are not conducive to mixing these two evenly. Therefore, extensive experimental experience in previous studies has led us to choose to add nerolidol below 50 celsius degree, new expression were given in our new submitted manuscript, please check it (Line 409-410).
Point 23: Line 454: by “the resulting cDNA” you mean purity or concentration or something different?
Response 23: Thanks for your kindly reminder, we mean concentration, and corrected the sentence to “The concentration of cDNA”, please check it in our new submitted manuscript (Line 480).
Point 24: Table S1: add the column with lenghth of product
Response 24: Thanks for your useful suggestions, we add it, please check it in our new submitted manuscript (Table S2).
Point 25: In whole text some editorial mistakes were made, I just wrote some e.g. line 19, 31, 101, 192, 311, 312, 370.
Response 25: Thanks for your kindly reminder, we have carefully reviewed the entire text, some errors have been corrected, such as Line 19, 32, 53, 90, 101, 103-105, 175, 192, 197, 208, 216, 230, 312, 314, 372, please check it in our new submitted manuscript.
